# A Qualitative Study Examining the Illness Narrative Master Plots of People with Head and Neck Cancer

**DOI:** 10.3390/bs9100110

**Published:** 2019-10-17

**Authors:** Kate Reid, Andrew Soundy

**Affiliations:** 1Therapy Services University Hospitals Birmingham, Birmingham B15 2TW, UK; 2School of Sport, Exercise and Rehabilitation Sciences, University of Birmingham, Birmingham and B15 2TT, UK; A.A.Soundy@bham.ac.uk

**Keywords:** cancer, qualitative, narrative, story, patient-centred care

## Abstract

**Background:** There is a need to understand the common plots (master plots) of illness narratives for people who are treated for cancer. Improved insight would enhance therapeutic relationships and help reduce stress for health care professionals (HCPs). **Aim:** Identify and refine the most supported narrative master plots, which convey meaning for the tellers’ lived experience from diagnosis to a year post-treatment for a group of Head and Neck Cancer (H&NC) patients. **Method:** A purposive sample of individuals with H&NC using a single qualitative interview was undertaken. A narrative analysis was used. **Results:** Eighteen people (57.8 years, six female and 12 male) with H&NC participated. The average time since treatment began was 10 months. Five master plots were identified: (1) The responsive and reflective narrative, (2) The frail narrative, (3) The recovery narrative, (4) The survive or die narrative and (5) The personal project narrative. **Discussion:** The identification of narrative master plots of people with H&NC enables HCPs to understand and prepare for the different stories and reactions presented to them. This is important to prevent people’s reactions being labelled in restrictive ways. The implications of recognising the different experiences are discussed further within the manuscript. Research is needed to build on these findings to promote better patient-centred care in practice.

## 1. Introduction

Head and Neck Cancer (H&NC) is the sixth most common cancer worldwide [1]. Approximately 550,000 patients are diagnosed with and approximately 300,000 deaths are caused by H&NC annually [2]. Research has identified a poor survival rate despite the search for new prognostic and predictive factors [3]. It is a collective term for cancers within the anatomical areas of the oral cavity, oral-pharynx, pharynx and larynx. The structures have intricate, coordinated movements, which are temporarily or permanently altered by the disease and treatments used. H&NC impacts on patients’ physical, emotional and social functioning in pervasive and subjective ways. The changes may disrupt eating, drinking, communicating [4,5], social interactions and mental and social well-being [6]. The experience may challenge a personal sense of self [7], increase levels of uncertainty [8] and negatively impact day-to-day life [9,10,11]. Despite such threats, the patient group has been reported as being less willing to request support [12,13] when compared with other patient groups. Research has also identified that psychosocial care of patients with H&NC cancer can be overlooked which may contribute to a lower standard of care [14].

Two initiatives in England have been designed to support the care of patients who have cancer; The National Cancer Survivorship Initiative [15] and the Recovery Package [16]. Both work streams emphasise a patient-centred approach and encourage self-management throughout care.

Central to the approaches is the therapeutic relationship between patients and health care professionals (HCPs), but high levels of stress for HCPs from increasing workloads and a low sense of control within their jobs [17] threaten to undermine the quality of the interaction and reduce the quality of care provided [18]. Poor interactions may also occur because of implicit biases made by the HCP team about patients [19]. With specific reference to H&NC patients’ poor interactions between them and HCPs can have significant and negative effects on psychosocial well-being [20].

Sharing and using illness narratives may be one way to develop the therapeutic relationship between people with H&NC and HCPs [21]. Illness narratives are stories that are expressed by individuals that allow the person to make sense of their experience. Recently, research considering illness narratives has identified the importance of the master plot within the HCP–patient interaction [22,23]. A master plot is a commonly recognisable story plot relating to the experience of illness. It has been identified that HCPs may judge and characterise the tellers of particular narrative master plots. For instance, HCPs can label plots with words like ‘unrealistic’, ‘not accepted’ or ‘in denial’ [24]. These words can infer a limited or negative characteristic of the teller and highlight the tellers’ ‘static’ response to the illness. Research has highlighted the importance of a broader understanding of master plots to include psychological adaptation, hope, and emotions found within the plot and to understand the relative nature of an illness narrative [23]. This illustrates the idea that individuals have a unique story that is told within a common or recognisable plot. 

The most frequently cited illness narrative master plots within the cancer include the Restitution (focus on a restoration to a pre-illness or pre-symptom status), the Quest (focus on an embracement of the present situation) and the Chaos narrative (an inability to tell a story because life at the present is considered as over). These particular plots provide details of quite extreme responses as detailed by a psycho-emotional adaptation and hope response [23]. These initial master plots require further consideration [21] and to the best of the authors knowledge no further development in the master plots of people with H&NC has been made. Understanding the master plots of people with H&NC will allow HCPs to have an opportunity to understand common reactions and consider them as known and expected plots. This is important because there is a need for HCPs to go beyond therapeutic emplotment (the way HCPs may structure information during interactions in order to instil specific hopes in treatment) [25], or a potential lack of support or regard for certain narrative master plots because of what they represent [24,26,27].

Given this, the aim of the current study was to identify common illness narrative master plots that are told by individuals with H&NC who have recently finished treatment. The objectives were to describe the plots, understand the psychology of the plots, identified characteristics of the teller, identify strengths and limitations of the plot and consider implications for the HCP–patient interactions. 

## 2. Materials and Methods

For the purpose of this research, a subtle realist paradigmatic stance was assumed [28]. This stance focuses on common realities experienced by individuals, in this case represent by common master plots. The results of such studies do not claim a single or sole truth from the knowledge created in research, rather a version of reality which is, to a more or less extent, relatable and recognisable by others. The methodology assumed for this approach was a hermeneutic phenomenology. This was undertaken using a single semi-structured interview. The interview was taken as part of a Q methodology (Q-sort) study. This section is presented according to the Consolidated Criteria for Reporting Qualitative Research (COREQ) [29].

### 2.1. Research Team and Reflexivity

The interviewer was a 45-year-old female who at the time of the interviews was a speech and language therapist working with patients diagnosed and treated for H&NC for a major UK National Health Service trust and a PhD candidate. The interviewer was not managing the participants; the participants were only made aware of the purpose of the study. The interviewer had received training in Q methodology and qualitative research for the purpose of this project. 

### 2.2. Setting and Context

The study recruited patients from a cancer centre based at a large teaching hospital located within the West Midlands of England. The primary investigator approached all individuals to invite them to rank 45 statements in relation to how much they agreed or disagreed with them. After the sort there was an opportunity to discuss through a semi-structured interview their experiences of the disease and treatment with the researcher.

All interviews were conducted in a quiet location on the hospital site, chosen because of its familiarity for patients, who attended outpatient appointments regularly. Other family members were not present during the interview. The primary researcher obtained demographical data prior to the Q-sort and interview. Clinical information relating to the tumour staging, treatments and time since diagnosis were obtained from the clinical record. 

### 2.3. Participants and Sample

A purposive sample was selected in order to establish common and pivotal stories, which the participants told throughout the statement sorts and discussion that this prompted. The intention is to seek a range of different participants so that a variety of difference viewpoints might be expressed. Individuals who had been diagnosed at least a year and treated with curative intent with head and neck cancer were approached at the cancer centre, part of a teaching hospital trust in the West Midlands of England. 

Individuals were included if: (a) They had at least a year since their initial diagnosis, (b) they had completed treatment which had an intent of being curative, (c) they were above 18 years of age, (d) they were able to understand English and (e) there was no clinical sign of a recurrent disease at the time of the study.

### 2.4. Method of Approach 

This study presents results from the semi-structured interviews that followed a Q-sort study [30,31]. The purpose of the Q-methodology study was to capture and facilitate the interpretation of experiences by the arrangement of statements, and then subject the statements to inverted factor analysis. Within a past study [30] participants with H&NC were required to rank 45 statements in order of most agreed with to least agreed with around their experiences from diagnosis through treatment and recovery of H&NC. The analysis identified the unity of participants’ voices across five themes; (1) meaning and attachment to illness; (2) overwhelmed by the cancer; (3) surviving or not; (4) change and recovery; and (5) keep control for the greater good of others. These distinct factors revealed an identification of common experiences, which were shared amongst participants and provided the basis to identify the common plot of stories that could be told by people with H&NC. 

The current study explores the interview data that was collected during the discussion that each participant had with the researcher after the Q-sort was completed. The semi-structured interviews after the Q-sort acted as reflective time which allowed participants to reveal and explain their choices and perceptions of the Q-sort statements in relationship to their experience of diagnosis and treatment. Participants were able to explain and define those statements that they most represented this provided a natural vehicle to reveal their own story and identify whether the factors identified previously [30] do indeed represent a plot of a story. Master plots or common stories reveal a similarity in the experiences of illness. The most common stories were focused on because the discussions would often revolve around the previously identified factors [30]. Participants were able to justify their choices by identifying how and why they agreed or disagreed with the statements made. In the process of this justification it became clear that the most important elements of the master plots would be revealed including; definition, the psychology behind the experience, the factors which may influence the experience and responses to interaction with HCPs. Transcripts of the interviews were completed by the interviewer within 48 h of the interviews. Non-verbal details associated with the discussion were integrated into the scripts. 

### 2.5. Sample Size

Eighteen participants completed the Q sort and had an opportunity to discuss the array of the statements which became the structure that the interview was based on. A purposive sample was selected to identify the central points of view about the experiences of people with H&NC. Sample size from the original Q-sort study [30] was based on obtaining one participant for every three statements, meaning at least 15 participants had to be recruited. The interviews were finished once the details of the identified master plots were considered saturated [32]. This occurred after 14 interviews and checked again following the final 4. 

### 2.6. Ethics

The study was conducted in a regional cancer centre at a teaching hospital. Ethical approval was granted via the National Health Service (08/H1202/96). Data were collected between July 2010 and October 2010. 

### 2.7. Analysis 

The analysis was designed to capture a number of narrative genres or master plots [33] through the development of a structural analysis [34]. Common and discrete experiences were identified through the Q-sort method. This method established significant differences in distribution of the phrases through inverted factor analysis [31]. The primary purpose of the analysis was to identify whether common narratives existed by using the previously identified factors as a starting point. This meant that an abductive and iterative process across 4 phases was undertaken as follows: 

(1) During this phase, the lead author used a priori knowledge from the previously identified 5 factors [30] as a basis to identify the most common master plots. The lead author immersed herself in the text and verbatim recorded text in order to establish the following information for each factor; the narrative itself, including the definition of it and what it represented for each participant and the characteristics of the teller and any responses that related to interactions with HCPs. Author AS acted as a critical friend to enable a broad understanding of each narrative to be initially identified. An example question from author AS included: Is there a plot evident from statements made and from the breakdown of information on it? If so, what is the main plot? Does the story have a clear beginning, middle and end? How does the story relate to loss? and how does it consider hope and psychological adaptation? What is the role of others in the story? What is the purpose of this genre? What does it reveal about the teller? These questions were guided by past research [34] that has identified the importance of understanding and detailing master plots as well as the psychology behind the narrative. The lead author presented answers to these questions for each master plot (based initially on a factor). 

(2) During this phase, the lead author refined the initial content and then provided details of each master plot according to the following aspects: (a) the goals of the master plot, (b) perspective of mortality and recurrence, (c) expressions relating to physical, psychological and spiritual well-being, (d) identification of psychological adaptation, recovery and hope and (e) characteristics as well as the over-riding themes of the story were examined. These concepts have been found to represent different narrative types from past research [35] and provide a way of examining whether these stories would fit into past types or represent new types.

(3) During this phase, the lead author undertook a further examination of the experiences given by participants. This phase documented the context to each narrative master plot. The lead author focused on detailing the view and experience of the treatment journey (identifying participants’ view of mortality, regrets, and decisions towards illness, progress and interactions with HCPs). 

(4) During the final phase, both authors considered several rounds of critical questions of each narrative that was developed and identified verbatim quotes to illustrate statements from each participant. These four primary stages can be obtained from the corresponding author.

For the presentation of these findings each participant is identified by two letters, followed by the page number of the transcript and the line numbers of the quote. A Appendix A provides an example of each stage identified above. 

### 2.8. Trustworthiness and Rigour

A critical friend, author AS was used through the process of analysis to question results, to develop the stages of analysis and to help create the narrative master plots. The number of individuals selected represented an adequate number for the purposes of information power [35]. The authors acknowledge that sample size is not something which methodologists necessarily seek to establish in Q methodology [36]. An audit trail of the analysis can be provided by the corresponding author for transparency of the process to be established. Reflexivity of the researchers is also provided.

## 3. Results

Twenty participants were identified and approached, and eighteen agreed to take part following recruitment. The reasons for not taking part included: not wanting to come to the hospital on a separate day from the review appointments but recognising that attending both an appointment and taking part in the research in the same day was too onerous and not wanting to take part in any research.

The sample included twelve male and six female participants; the average age was 57.8 years with an age range of 37–77 years. Four patients had surgery only; two had chemo-radiotherapy; and twelve had multimodality treatment (seven had surgery followed by radiotherapy; and five had surgery plus chemo-radiotherapy). The time interval since the last treatment for primary disease H&NC was between six months and eighteen months, with an average time interval of ten months. The total length of the treatment period for each patient ranged from zero to 167 days. Table 1 provides details of the participants.

### 3.1. Illness Narratives

Five illness narrative master plots were identified. Table 2 and Table 3 provide details of the narratives and their defining features. The main results are presented as illness narratives to describe: (i) each plot and narrative; (ii) the interactions associated with the teller; and (iii) the adaptations and coping mechanisms of the teller. 

#### 3.1.1. The Responsible and Reflective Narrative 

This narrative was the most common, with seven (7/18, 39%) people identified as presenting it. The basic plot to this story is that life is fragile, and decline or changes need to be managed by the teller and will be. The tellers are likely to keep the real impact of the disease and treatment to a limited number of people. Life is portrayed as being challenging and is expressed in objective rather than emotive terms. Previous experiences the tellers or their family have had are referred to during the current situation and act as examples they recognise of people they know well coping with difficult circumstances. A pragmatic, fatalistic approach to storytelling is used as they express the suffering, loss and mortality and relate it to their current experiences. 


*“I think it’s our upbringing. My mum was never very demonstrative; we grew up in the Cuban Missile Crisis. One day as we went off to school she really hugged us. I understood much later she was so worried as we went off to school that the end of the world was going to happen ..so as a teenager its very formative it’s that blitz mentality.”*
*CF: 3.7–3.15 (Participant ID: page number line start–line end)*

Scenarios relating to the illness are communicated in a logical and sequential way so that information believed salient by the teller is described in eloquent ways to the listener. The plot suggests tellers have considered their own mortality prior to the current diagnosis. One individual stated in response to the statement “it’s difficult to think of your own death”: 


*“Of course it’s ridiculous (not to contemplate your own mortality) you have to think about that anyway we are all going to get there. I’ve always been a realist I always thought if anything should happen to me this is what I want as opposed to I won’t think because it will tempt fate.”*
*KG: 9.18–9.28*

The teller appears to seek normality from pre-morbid activities, including work. Although this might be in a different format, structure and familiarity is critical as a way of adjusting to lived and possible changes. To enable this, the teller is receptive to any treatment and the desire to be treated is always expressed. 

The news of the diagnosis was a shock and the treatments challenging but being cured of the cancer is the most important aspect of treatment for the individual. Individuals express no regrets around having opted for treatment. 


*“When I was diagnosed it just went bang in my head, but you have to get on with it.”*
*CF: 1.1–1.2*

The ultimate goal within this plot is to be as well as possible despite the challenges that both the diagnosis and treatments present. 

The detached position of this story means the teller is able to recognise that individuals react differently to a given situation. The teller understands the importance of experiences being relative to an individual’s situation and circumstances. For instance, one individual stated: 


*“Everyone copes with it in their own way, drama queens versus bluffers.”*
*CF: 2.40*

The repercussions of recognising the individuality is shown by the teller of this story not being presumptive about their experiences of cancer being similar to others. They might, if asked about the treatments and the experience, discuss it with people but there is some reticence about what to impart and the level of detail to discuss because they recognise the differences. 

##### How the Story Uses Interaction and Treatment

The tellers of this plot, through choice, appear self-contained and the story suggests that privacy is sought. The changes experienced do not have to be understood by others because this is not necessary for the teller. There is an acceptance within the narrative that the impact of their diagnosis on people who they are not close to will be superficial. They are pragmatic about this, noting they would have reacted in the same way if they had heard of an acquaintance that had been diagnosed with cancer. There is no desire to be the centre of attention following the initial diagnosis. Acquaintances or work colleagues are told limited amounts of information from the premise that few people need to have all the details because it does not concern them. The tellers of the narrative will have emotions but they do not want to risk having to handle others’ emotions when they are handling their own reactions to their current situation.


*“I didn’t want anyone to know....it was too scary it was in me....but since I left work (on sick leave)….my family knew and then I came home and I told my line manager I don’t want everyone to know; I was just going to be off.”*
*MG: 1.12–1.19*

These tellers use skilled communication techniques in order to help make judgements about the HCPs they meet. There is recognition that HCPs have advanced technical skills and knowledge. Relationships are only built with those perceived by the teller to be beneficial and the details of the conversation are remembered. 

The tellers of this narrative never felt alone recognising that the relationships created with HCPs contributed to a sense of belonging. There is an appreciation of the tasks HCPs carry out and a sense of pragmatism associated with the information that needs to be given or kept to a minimum during those tasks.


*“When they took a neck drain out and they said it wasn’t going to hurt, it did, but what are they supposed to say “This is going to hurt a great deal so just tense up even more and watch how painful it can be.”*
*CF: 1.31–1.36*

It is unlikely external information from either the internet or written literature will be referred to because there is a perception that the details are too general. Instead, a reliance on chosen HCPs to discuss the intricacies is favoured. One person commented:


*“Your imagination is running riot and what you need is the parameter that says “but that’s not even on the cards because you aren’t in that ballpark”. It’s so helpful. You don’t know and you need to talk to someone who can say “hang on let’s go with the reality here”. If you read on websites you don’t know how much relates to you …..you need that quality input, guidance from what to expect now or in the next few steps.”*
*KG: 6.17–7.12*

The tellers define for themselves the direction of recovery and the parameters they want to make judgements on. Chosen HCPs are also sought to support and understand the trajectory and rate of recovery. The story does not want false hope from HCPs. The tellers are pragmatic and have an acceptance that HCPs cannot impart only good news. They know it is possible for a recurrence to be discovered. In part, the responsible, reflective behaviours within the narrative demonstrate that the decline from the disease and treatment will be responded to by the tellers. 


*“You might not want to believe it but that’s different from not having it explained to you.”*
*CF:1.20–21*

##### Psychological Adaptation and Coping 

There are different methods of coping detailed by the tellers of this plot.

Acceptance and a degree of self-containment are demonstrated through acknowledging that the diagnosis brings about changes in function. Thus, the teller embraces a forward-looking stance that is ready to be adapted to deal with the real or potential challenges faced.

These methods focus on solutions, forward looking and acknowledging their own mortality. This acknowledgement allows specific strategies to be maximised (use of good time management, problem-solving, and reframing, upward and downward social comparisons) but minimise reflection on loss and well-being prior to the current diagnosis and treatment.

In this story no treatment is off-limits as long as it gives the maximum possible outcome. Details relating to the diagnosis and treatment are described in the narrative and compared with previous difficult experiences for themselves or others. This comparison provides a wealth of coping methods that might be considered. This facilitates the embracement of the situation.

There is insight that the tellers, or others close to them, have suffered previously and the events with associated emotions are referenced but presented as transient. The fluidity exists because of an innate confidence in their ability to respond to the changes brought about by the disease and treatment in order that life, in their terms, can continue to be worthwhile.

There is evidence of resilience with a sense of reality and humour in descriptions of symptoms, which is part of the coping style.


*“Your mind obliterates the horror. It was unpleasant of course it was. I remember when I got diagnosed suddenly the cast of Ben Hurr appeared- the MacMillan Nurse was there and others –well come on you have to be daft not to pick up on it?.”*
*CF: 2.19–2.24*

The experience was coped with by noticing the differences and knowing how that made them feel. 


*“It was the lack of control I had, I felt I had no time in the beginning. I had to go to so many appointments I knew it wasn’t good news and it does change you perception of things.”*
*CF: 3.1–3.5*

The tellers of this story believe they have recovered more slowly than they had expected prior to treatment, which they found frustrating. 

There is both hope and belief that the treatment will have enabled survival but there is recognition that there are other potential outcomes; importantly there is an acceptance of the worst-case scenario, a further cancer being found. The way to cope with the concern of a recurrence for the tellers of this narrative is to understand the range of possibilities, use fully coping techniques and to attend appointments in order that HCPs are given the opportunity to assess the symptoms and investigate if any further treatments are possible. 


*“You see the surgeon, whatever will be will be; if it’s going to come back seeing the team or not seeing the team won’t stop it but seeing them might mean something can be done about it.”*
*MH:2.15–2.16*

#### 3.1.2. The Frail Narrative

This narrative was identified by two (2/18, 11%) people identified as presenting it. The basic plot to this story is that the physical symptoms and the emotional impact of the disease and treatments have taken their toll. The tellers of this story demonstrate that their energies are precious and finite. They have previous experiences in their lives where if something does go wrong for them, their perception is that it did. They are wary of depleting their energy so only engage in what they judge to be the essential aspects of life. 

This story revolves around unremitting symptoms, and negative emotions associated with social interactions, which are recalled by the tellers if opportunities arise to describe them.


*“It’s very hard on him (husband) he tends to laugh in situations that are really not funny…he more or less said yesterday “I’m glad it’s you not me ha ha” …it’s difficult.”*
*AB:1.20–1.24*

The symptoms for these tellers appear to be at the forefront and difficult to be distracted from their impact, which means they perceive themselves as more irritable. The irritation is exacerbated for them by the continued interruption to their lives from the cancer treatment. 

Unlike the other plots who do not describe any sense of being stared at by people, this group allude to an altered body image and a reaction from others, which they respond to again with irritation.


*“Like the weight loss I’ve lost a stone, don’t know why—I’m having my supplements but I can handle the weight loss as long as I don’t lose any more..If I have seen people they say “blimey you’ve lost a lot of weight” and I tell them (the reason) and that shuts them up.”*
*GH:4.31–4.37*

These symptoms drain their limited energy and resources and explain the desire to be distanced from people and social occasions. The tellers of this plot identify with being disabled rather than ill. There is awareness that whilst the disability might not be visible, the limitations to vital functions restrict fundamental aspects of their life. The teller needs to recount these restrictions as part of the explanation as to how they are unable to function at the moment.


*“I have got out of the social network and I have not wanted to go out and have a drink yet. But I will. I can use the crème fraiche as the fire-extinguisher because the next day I seem to suffer.”*
*GH: 4.7–4.10*

The tellers of this story are the only group that admit there are times when there is regret in having chosen to have treatment. The other groups refute this sense very strongly. For the tellers of this story at the point of diagnosis, not being treated had been considered as a real possibility and therefore it is a real choice that they feel able to regret and wonder at the alternative no treatment. 

They are frustrated by systems and processes, which although intrinsic to the running of health care are seen as further threats on their ability to manage. They have limited resources in which to engage with health care and the process of engagement adds to a sense of a mere existence. 

##### How the Story Uses Interaction and Treatment

There is little impetus to provide explanations to friends on the periphery of the recovery, because this is judged to be a drain on their limited resources. Depleted reserves are guarded and barriers are put in place to reduce the potential challenges to the current way of living. 

One of the tellers for this plot described a sense of reviewing his previous life experiences to explore reasons as to how he deserved the current situation. 


*“I started to go back over things from a long time ago-What have I done to deserve this? I had this one memory of my mum; my dad took me to a hospital window and she waved at me I was five and then I never saw her again and then three years later I found my dad dying I found him actually dying.”*
*GH: 3.20–3.26*

The teller of this narrative expresses no intention of being more engaged with their social network because they cannot see beyond the current symptoms. The situation prevents them from integrating and they are less confident within their social network than they were previously. One individual commented:


*“My partner says “what are you going to do not going back to work?” and I think “don’t start I want to recharge.”*
*GH:5.31–5.33*

There is a tolerated level of interaction within their social group, but it is monitored and will not continue if it is believed to be too much for their reserves, even if there is pressure from close family or from work to opt back into routines.


*“I just want to be left alone that’s why when I was at work I would wonder why people were putting me through hassle, I thought ‘don’t you understand why do you keep going with this?’ I just want to opt out and I can get my pension.”*
*GH:3.6–3.10*

Discussing key events and consequences is exhausting partly because of a sense of being misunderstood and the need to justify the subsequent limited involvement in life. Retelling their own story becomes a task in itself and one open to misunderstanding. The impact of this is that discussions become minimal with family members because the tellers of this narrative have a preference to communicate with HCPs. 

The HCP team are considered to be empathic and trusted to judge correctly the length and specific content of consultations. Prior to appointments written information will be read and used as a basis from which to discuss the specifics for the individual. Meagre energy reserves are used-up, which means detailed discussions are hard to remember. To minimise this effect the teller is likely to have prepared questions, and will refer to them, using the structure created to aid their recollection of what is said. The interaction with the HCP for the teller is a source of hope, because they are going against all other principles. The hope being that this interaction can change their life or influence it therefore with limited or very small amounts of energy they will invest in time with HCPs and be expectant.

##### Psychological Adaptation and Coping 

There is no appreciation by the teller of how long it would take to recover from the definitive treatment. Energies remain concentrated on the endurance of day-to-day activities, which means that there are few reserves to plan for the future. Such beliefs are manifested through not wanting to think too far ahead to family celebrations or days away from the routine, which will include visits associated with their health care needs. At a reduced pace of life there is an attempt to cope with the personal isolation, in the knowledge that others cannot understand the true ramifications. One of the tellers stated: 


*“Unless you’ve been through it that battle of emotions they just can’t appreciate the ups and downs. I just feel very vulnerable- I’ve given up work……..no one can understand what’s happened to me ….I don’t think anyone, unless you experience it could know what it actually does to you mentally or physically.”*
*GH: 2.33–2.38*

There is no evidence that they have learnt or reflected from previous challenges in their lives. For the tellers of this narrative, previous experiences have left reserves depleted and the current circumstances reduce energy reserves further. Thus, life is spent focused on the present, managing the present ability to make it through a day within the limitations of reserves. If, within daily activities, other people are observed by the tellers of this narrative to be buoyant, despite tangible signs that life should be difficult, the tellers wonder whether they are portraying a false reality and not being truthful about how they really feel. There is no belief that the tellers can emulate such examples, as they are not inspired by such examples to either behave or think differently.

Cure remains the ultimate goal and includes the restoration to former self, which has not been achieved but is hoped for in adversity. There is a contrast between where they want to be and there is a reliance on the medical system or professionals to achieve that. Whilst waiting, there is an acknowledgement that the best they can do is manage to get by on a day-by-day basis. There is restlessness with the current situation and an ongoing search for the former self, which the teller believes is still attainable. One individual described the situation:


*“I can’t get used to not being active, I don’t think I have adjusted to the physical changes.”*
*GH:1.7–1.8*

Symptoms that might be a recurrence are of concern, but HCPs are trusted to manage the reality through appointments and their knowledge of the individual’s case presentation. The sense of having minimal reserves means that the tellers of this story will not focus on the possibility of a recurrence. The teller is passive in their circumstances and reliant on HCPs who they report very strongly always treated them as an individual.

The tellers are ambivalent at the concept of being ill but their lives are limited severely by their ongoing and, for them, overwhelming symptoms which bring them into a cycle of just about coping. They have a generalised sense of hoping that medicine will help, which is why it is so important for them to attend clinic appointments. 

#### 3.1.3. The Recovery Narrative

This narrative was identified by three (3/18, 17%) people identified as presenting it. The basic plot to this story is that one can understand and respond to the reality of the illness in an effective way, which is helped by having reliance on others. There is a real sense of wanting to progress with the treatment without the need to know much about the specifics or reference to past experiences. Knowing that something could be done and being cured are strong themes for this plot. There is evidence of being stoical and resolute through what are described as “horrendous treatments”.

The tellers of the story are openly dependent on key HCPs and recognise there are also key members of their family that enable them to integrate socially. They will seek support from these crucial people.

Whilst they know they are not able to participate like others in social situations, they actively choose to share an activity with their families, which they welcome and appreciate. Social integration is helped for the tellers of this plot because there is no sense that other people stare at them, which may explain how they feel able to integrate within their communities.


*“Going out for a meal I have the soup of the day no roll. The family tuck into a full meal and I try bits off their plates. Part of me would love to have what they have. There’s enough people who can cover for me and my daughter is very protective.”*
*GM:1.20–1.26*

The story is defined by taking action in order to continue life and the desired pleasures from social interactions. Definite differences are noticed within the interactions and activities, which continue but differently.

There is openness to life not being the same but to do nothing is never an option for the tellers of this story who are proactive. There is little focus on outcome or a similar end point for patients, which is seen to be part of HCPs’ remit rather than something for them to consider in detail. The possibility of recurrence is not considered because this group is more focused on the current situation from the original premise that they would not be diagnosed with cancer

There is an emphasis on what can be changed now in order to manage in the future more easily. The tellers of this narrative are much more in the present and will modify their current behaviours and later reflect on the future in the hope that their skills will improve but from the reference point of the current skills.


*“I went to the pantomime with my son and is two children. I really enjoyed it. I got my water and there were lovely toilets…I can do some of this I thought………My son bought me a cup of coffee with extra milk and we sat in the restaurant and I really enjoyed returning to normal life….it’s just when will go to a restaurant and ever have a proper meal..?.”*
*GM: 3.6–3.16*

This story embraces change and loss and is accepting of the diagnosis. The tellers always felt it was possible to have been diagnosed with cancer and would recount close members of their family having died of the disease. However, they found it hard to contemplate their own death and had never thought about their own mortality prior to this diagnosis.


*“I always thought it could happen to me. Both my parents died of cancer so I’d be stupid not to think that after I’ve had the CT scan.”*
*EN: 2.13–2.14*

##### How the Story Uses Interaction and Treatment

The teller of this story often uses interaction and time spent during tasks with HCPs to enhance and develop trusted relationships. Where they perceive that HCPs have taken the time with them, they value the interaction in the moment and are likely to listen intently, preferring discussion to reading any information provided, which is rarely looked at. One individual commented: 


*“So of a team of about 90 people about four or five (are beneficial) and, you can narrow that down further to those who had the most impact. …. Some people are just filling in their forms- it’s a process “I’ve done my bit” –like a sausage factory……… if only you could be seen by 30 people in a better way than 90 doing a half-baked job of it.”*
*EN:9.35–10.3*

They would remember discussions with the team who they felt treated them as individuals. They did not write anything down or read the literature; instead, they relied on building relationships with key people to create the conversations. 


*“I never read any of the information. I guess I didn’t want to know…my wife read up on it. I just did not want to know. It was as if the detail did not bother me. I wanted to know a little bit but when they give you a book that thick on neck dissection (demonstrating with finger and thumb an inch) that was of no interest. I left it in the boot of the car.”*
*JH: 1.1–1.7*

There appears to be a need for support from others, which is vital, as they do not believe they have to be self-sufficient. The danger of the desire to try and build relationships appears to be that they are sensitive if an interaction is judged by them to lack a standard that they have come to expect. One individual stated when having an ultra-sound guided biopsy: 


*“At one point there were three, four, five people in a small room—it’s very enclosed. That’s when you feel alone you feel like saying “Hello I’m here. Talk to me!.”*
*EN: 4.29–4.31*

Further to this sensitivity to non-verbal situations, it appeared to be that they may not question the advice or information given by HCPs or seek a deeper understanding during consultations. There are limits as to how much they might remember and it is not easy for them to build a complete picture of the current situation. There is an expectation that HCPs will repeat information for them and, whilst information might not become misunderstood, repetition reminds them of the reality of how long treatment effects are. 

They rely on HCPs to make the decisions or choices and they are less likely to be opinionated than other stories. There is evidence too that they can forgive behaviours by HCPs because of the role they have in decision making and their care. 


*“We hung on every word my partner, my sister my brother, we are from the old school, the consultants are God and you bow down to everything they say. The surgeon was ultra, ultra-professional ..bit distant really—Is that their job, to be distant, so that other people can be more touchy feely or whatever?.”*
*EN:9.4–9.10*

Information will be understood through discussion during interactions with HCPs but more so when they are experiencing it. The danger of such an approach was that that it is not possible to assimilate the information with the discussion until the symptoms are a reality and then there can be a sense of dissatisfaction. This group were the most dissatisfied with the rate of recovery. One individual commented: 


*“No, [current presentation] it’s nothing like the reality the long-term effects are not particularly explained the fact I am numb from the top of my ear, puffy round the neck ……….that was never really explained…….”*
*JH:1.10–1.14*

##### Psychological Adaptation and Coping

The embracement of change appears to be accompanied by the need to act and utilise past coping skills. These skills are used to create a framework through which specific goals are achieved in order to have a sense of normality. There is a need to achieve normality by focussing on future activities or milestones such as leaving hospital, visiting a friend or getting back to work. The tellers of this story seek normality and are less accepting of the need to adapt. One participant stated: 


*“The last time I was in hospital was 18th October and I thought there was a possibility of going back to work before Easter in the April. Even then … I thought once I’m off that (Naso-gastric) tube I’ll be up and away and off I’ll go; ……(shakes head) and the first meeting I had about going back to work was after Easter.”*
*EN: 1.34–1.40*

The teller does not deny the difficulties faced and through discussion is aware of how much longer recovery has been. They state that the way of coping has been to be task orientated with smaller goals that are achievable or by necessity approaching activities in different ways but they will still be participating in them. 

The tellers identified strongly with being ill and were surprised at their abilities to adjust physically. They acknowledged the treatment undergone and described it emotively as “horrible” and “horrendous”, leading to “utter exhaustion.” The treatment situation was managed by breaking down processes into smaller chunks. 


*“You deal with it in the bits that you can. For the radiotherapy I got the timing sussed. I knew the different sounds to listen for …this sound then that sound and I could work out how long ‘til it (the radiotherapy machine) stopped.”*
*EN:12.23–12.27*

To manage the symptoms and effects of the illness is to overcome and continue life. The symptoms and problems created by the illness are viewed as a hurdle, examined for what they are, but can be beaten. 

The teller is able to recognise times of vulnerability and express them in emotional terms, but still there is an emphasis on an ability to overcome the symptoms. One participant stated: 


*“We both dreaded (patient and partner) the weekend at home. When I had the feeding tube (Naso Gastric Tube) in, it was horrendous we would ‘phone the ward, they were brilliant and whoever was the duty-registrar would say “Bring her in” even if it was only for two days….We felt very vulnerable particularly when it blocked I tried with fortisips (feeding tube supplies) and during this time the weight just fell-off me. I never noticed …….and you are not wearing the same clothes. You became so vulnerable to a little tube blocking up…that’s the only time we read the information— tricks to unblock the tube …(laughs) the one time …….Halleluiah.”*
*EN: 4.1–4.32*

The teller can see a point in the future where, though challenging, recovery will have been achieved. The action taken is never doubted because to have misgivings would be to look backwards—something that this particular story denies. 

#### 3.1.4. The Survive or Die Narrative

This narrative was identified by three (3/18, 17%) people identified as presenting it. The basic plot to this story is that the teller is focussed on living as well as they can. There is a sense of the teller being an optimist and lucky to be as well as they are. The tellers judge themselves as impulsive, and they are not limited in the activities they participate in. The teller appears to detach themselves from being a ‘cancer patient’, even to the point where they are passive during interactions. It represents an idea that ‘I am me, because I continue to function and live how I wish’. Interactions and activities remain focussed in the present and rarely move to think about the future.

The way they lived before the diagnosis is perceived as ended and they do not allow past stories to be continued. The teller acknowledges changes from their pre-treatment selves physically and that there are implications on how they live their lives (this included for one person a permanent tracheostomy). They are comfortable with their personal values, which creates a sense of a stable self and this is reassuring to them. 

The teller of this story seems passive towards taking action relating to their circumstances. Passivity is demonstrated through not searching for information on treatments or reading literature given to them and not being active during their care either when in hospital, or when discharged. There is a sense of fatalism and little point in worrying about what is not within their control. The scripts offered little evidence of individuals personally adapting to enable daily requirements or different skills to be carried out.

##### How the Story Uses Interaction and Treatment

There is no evidence of being exhausted physically and some social activities continue. The tellers do not describe pain as a key symptom and they are, they believe, able to be spontaneous. The ability to live and thrive in the present circumstances was often associated with an acknowledgment of a reliance on others. 


*“I didn’t have to think about it I just thought go ahead whatever you (surgeon) need to do.”*
*SN 4.20–4.21*

They acknowledge that it is the prerogative of acquaintances not to relate to them after treatment and they do not proactively try to re-engage them. The new limitations to their social network are however noticed. 


*“You do just have to get on with it. I felt isolated to start with and then it got easier, but people I have known all my life you look at them and they have no idea and you think…..(looks into middle space) well it’s their choice.”*
*SN: 2.3–2.5*

The tellers are likely to seek and receive support from close family. There is an appreciation by them of support through the treatments they have had.


*“I think it gives you a lot by knowing that there is someone really close and they are next to you and going through it.. doing everything you are.”*
*CA 4.13–4.15*

During consultations their family or close friends are likely to be involved in discussions. The tellers acknowledge this is helpful because they are unlikely to remember the specifics of consultations.

They want honesty from the HCPs, which they describe as straight talking. They do not read written information and prefer to experience rather than talk about prospective treatment effects. They live a day at a time. They give little evidence of processing information until it is experienced. An extreme example of this is illustrated by a participant who disclosed that for him there was no real sense of understanding the reality of an altered airway from meeting a similar patient pre-treatment. The individual stated:


*“I didn’t understand what happened to me even when I spoke to a guy who had had it; (a laryngectomy) it went over my head, when I left the appointment I didn’t know where I was. I met him and it made no sense.”*
*SN: 1.11–1.13*

This group will seek the opinions of HCPs and will defer important decisions to them, noting that the health care team have the knowledge that they do not. One individual stated: 


*“I don’t know if it’s cured or what ……..the doctors and the nurses know better than me.”*
*SN:1.28–1.29*

##### Psychological Adaptation and Coping

Treatment was seen as being the only option that could be taken but understanding the reality of what it really entailed was poorly understood until experienced. Adapting to treatment side-effects within the narrative is not evident. The reality of the present dominates and acts as a distraction. The tellers of this story do not experience ongoing pain and do not believe that they are prevented from being spontaneous. There is no regret in having chosen to have treatment and the consequences of the situation are faced as the circumstances require. Their thoughts and actions are not preparing them for significant deterioration. They want to deny it or avoid thinking about the future. 

One individual commented:


*“I know I had to have the treatment, I didn’t want to die so no choice….but awkward to think about.”*
*Sort number 7: 2.23–2.24*

Another reflected:


*“All I wanted to do was get it out of my body…all I could think was “no I just want it out of my body.”*
*Sort number 11: 3.16–3.20*

The tellers of this story judge that the speed of recovery has been good, but the total length of recovery was still lengthy. The teller often give examples of being reliant on others such as close family members. This reliance may reduce the need for them to adapt or to learn new skills. One individual commented having had a laryngectomy: 


*“I’ve never used a computer, can’t use the ‘phone because it feels like a heavy breather but who do I text ……….what if I need help and the wife isn’t around?.”*
*SN:2.14–2.16*

The diagnosis of a life-threatening disease has prompted a discussion about their mortality within their family. The discussion has included acknowledgment of the seriousness of the disease and financial matters have been attended to.

This group accept their current function as being very lucky and can describe a life beyond the treatments but will express a fear that a recurrence is possible. The sense of being lucky is threatened however by any symptoms that are experienced because this might be a further cancer. The effect of the symptoms on the teller is that they become really anxious because their luck is changing despite the involvement of the expert team. There is often an appreciation for the present circumstances and they tend to view facts optimistically, believing that luck or external factors have played a part in their recovery from treatment. The optimism is supported by what they see as a quick recovery which they are pleased about. However, the optimism is precarious, and can be knocked if they learn of other people’s health deteriorating. One teller confided that they were concerned about their own possibility of a recurrence because they had learnt of someone that they knew having further disease. 


*“It’s always a fear in the back of your mind. You hear of people and I talk to people that have cancer and it’s come back and I think will that be me—it’s a fear I have.”*
*CA:1.12–1.16*

#### 3.1.5. The Personal Project Narrative

This narrative was identified by two (2/18, 11%) people identified as presenting it. The basic plot to this story is of a project controlled by the teller who is at the centre of treatment throughout—independence and control are vital. This narrative type will reveal details of ‘a good recovery’, this is often evidenced by secondary gain that are stated. Altruism is a driving force. The tellers have a sense of being liberated, and able to make judgements about HCPs and health care systems. They are opinionated about what is important for them as they continue with their life. The tellers are active in making choices and act on them; they have embraced what has happened.

They have a renewed sense of purpose, developed through the experience, which may mean they will pass comment, or offer solutions for what they perceive to be difficult processes that they have had to engage with. The experiences have changed them, and they believe their lives have become more meaningful. The tellers’ past appears to have less value, since the narrative will often focus on the present and how much more fulfilling their current life has now become, as they describe secondary gain. One individual stated:


*“I have had 40 years of being a good corporate citizen somewhat repressed…but now I do not need to deal with the mundane and idiosyncratic parts of the corporate world.”*
*AS:2.8–2.12*

##### How the Story Uses Interaction and Treatment

The tellers of this narrative perceived themselves as being key members of their family and believed that what had happened to them upset others close to them.

Consultations with HCPs enabled specific literature, which would have been read prior to the appointment to be explored further. The tellers did not feel strongly that they were treated as individuals by the team. They were, they believed, able to judge whether individuals in the team were competent and dismissive of those unable to discuss in depth the details they wanted to discuss. There was a sense that HCPs earnt their respect rather than this being implicit from their role or reputation. Whilst in hospital, physical limitations would not stop them from discussing their views of care with the team. One individual stated:


*“I had a stand-up row with the (ward) sister via my pen and paper….I had a row using capital letters—Something like the PEG is so simple. “What else can’t you do if you can’t do that bit?” I never saw her again—that’s a training issue.”*
*AS:4.22–4.39*

The tellers of this narrative will have carried out comprehensive research of their disease and management and they will use technical language during discussions with HCPs. Having a grasp of and familiarity with the topic appeared to ease their acceptance of the situation. The tellers aimed to understand every intricacy of the treatment and processes because of a belief that they would recover and wanted to have an explanation that they could understand completely. Discussion through which they would use technical jargon with their team about their expectations of recovery would be expected. Familiarity of the topic for them enabled complex ideas to be considered beyond those of survival and would include predicted or likely quality of the life post-treatment.

##### Psychological Adaptation and Coping

The tellers appear proud and amazed that they have, they believe, coped well and do not consider that they have been ill at diagnosis or during treatment. The teller will often identify that experience had not been easy and that treatment used a lot of energy. They will also often identify that re-appraisal of their life was part of the process. 

One individual reflected:


*“I never thought of myself as ill. Going in or coming out I limped around couldn’t open my mouth properly- all the accoutrements of illness- but without feeling particularly ill; so it’s not like I have a long-term disease even though I can still visualize a fixed chunk of time as opposed to a chronic condition…some people do define themselves by their illness. This does not define me at all, my values define me and none of those include being ill.”*
*AS:2.31–3.9*

The tellers of this plot were independent in routines of their care post-treatment. During in-patient stays, an evident sense of vulnerability was felt, primarily from a perspective of placing their lives in the hands of HCPs. Relinquishing control was difficult for them. One of the tellers recounted:


*“Coming out of hospital was frightening but I was equally frightened in hospital, because you had no control over what happened to you so you were totally reliant on other people you put your life literally in someone else’s hands.”*
*KK:1.31–1.35*

The tellers of this story had grown in confidence and have a degree of self-admiration for the way they have responded during their recovery. There are no regrets about treatment and no doubts around being treated. The tellers of this narrative had never believed it was possible to have been diagnosed with cancer and found it difficult to contemplate their personal mortality. Cure was not the ultimate goal for the tellers of this narrative to consider, because the quality and detail beyond survival was important too.


*“Once I knew something could be done I wanted to know all the ins and outs of every little detail of all the jargon. I wanted to know on the assumption I would get over the treatment I wanted to know how I would be. Cure was not the only issue it was the quality of life ….not just the length of life –not at all costs not if I was going to be like a cabbage.”*
*KK:1.13–1.22*

The tellers perceived that the treatment time passed quickly, but recovery was still judged as protracted. They recognised that recovery for patients was individual and effected by patients’ reaction to the circumstances they are in. One individual commented:


*“The symptoms and the journey might be similar even if the outcomes are different the stages might vary according to their (patients’) personality. ……It’s hard to generalise.”*
*KK:3.34–3.37*

Symptoms were researched generally in order to inform specific and more detailed discussion with the team. Having such conversations increased the tellers’ insight and ability to describe changes or further challenges during recovery. The tellers expected a high level of detail to be discussed within a consultation by the HCPs about their recovery and how this might be demonstrated. If outcome measures could not be applied, there was still a desire to describe for the team the changes they could recognise. There was also the belief by the tellers that by relating their experiences, services would be improved for future patients. 

## 4. Discussion

The results describe the participants’ experiences beyond the biomedical details of the disease and recovery. The study critically advances the consideration of alternative narrative master plots to the most common identified within H&NC [26]. This understanding provides an identification of participant attributes and experiences that can be easily understood but often overlooked by HCPs [22]. These narrative master plots reveal the current beliefs, circumstances, psycho-emotional adaptation and hope that patients may have during treatment for H&NC. Further work needs to establish what factors may influence the choice of narrative master plot and how plots may change. In this study, there was variation in the role that participants believed HCPs had. 

Storytelling has been identified as an effective intervention for improving the emotional and mental well-being of people with cancer [20]. The most important consideration from these findings is to recognize the importance of hearing a participant’s story and moving away from a directive approach towards an indirective one during clinical interactions. Time should be allowed within an interaction to explore the individual’s narrative so that what matters for the individual can be stated rather than presumed or implied. In a health care system, targets have to be achieved and contacts justified and it is possible that tasks are completed at the expense of therapeutic interactions because HCPs become too busy to notice what might be important to patients at an individual level. If HCPs were to become more alert to these common narratives within their work setting and avoid the temptation to categorise or stereotype the people in their care, discussions might become more holistic and patient centred. Categorisation of narrative master plots have been identified previously when HCPs hear particular illness narrative master plots [26]. Categories are often limited to a few short-hand expressions relating to psychological adaptation, which become abbreviations of the reality a HCP may state that an aspect of psychological adaptation is limited, “*the patient is in denial of what is happening*”, or that a patient has misplaced hope, “*the individual is being* unrealistic”. These categories do not represent master plots fully and HCPs may require greater insight into how the master plot represents the psycho-emotional adaptation of the patient. The other danger of identifying a participant’s story is that it is seen as a way of searching for a category that might best fit with rehabilitation. Once identified, HCPs may perceive a need to change or ‘fix’ the patient’s story and use a directive communication approach to achieve this, e.g., a HCP may say “*what we should concentrate on are the physical symptoms we can improve (increasing oral intake, reducing reliance on pain medication) rather than on some of the uncertainties”*. Such a comment would be in contrast to a need to understand and work with the narrative presented in which a patient might express their fear or frustration because of their need to express their view of their current situation [21,23]. The value of patients being listened to is that the HCP–participant interactions are more effective. A major reason for this is because patients feel understood and respected for their current perspective. Listening to stories allows HCPs to become more aware of environmental, psychological and interactional factors, which might influence a patient’s shared expression [37,38,39]. Indeed, patients are more likely to trust HCPs and disclose what matters to them if they believe that the detail that they describe is being listened to in ways that are empathic. It is important to note the idea that narratives are on ‘moving ground’ [40]. This means there is a need to take time to interpret the structure of the stories in the context of the culture in which the individual lives and understand the relative importance of any story presented to an HCP. An interaction that can foster the sharing of narratives can enable joint action and encourage positive attitudes and behaviours from individuals with H&NC.

The ability for the narrative master plots to change is likely to be dependent on several factors including [22]: (a) the paradox of chronic illness [41], which states that individuals are impelled to simultaneously accept limitations because of the constraints imposed by the illness and defy the limitations to realise greater possibilities of living with the illness. This implies that one dominant story type may not be the only story identified with by an individual with H&NC (b) an opportunity for reflective time on loss and/or an environment that encourages sharing the narrative may help plots change and become more structured. This could be from a chaos narrative that identifies no ability to accept what has happened and no plot structure to a sad narrative that emphasis loss and has acceptance of loss as part of the plot. (c) A reconsideration of how hope, purpose and meaning is defined which could help the individual reconsider the importance placed on a medical cure. This may be best achieved by observing other stories and considering whether the plots and content could be something an individual could also tell. And (d) HCPs may assist interactions by identifying self-reflective empathic questions (e.g., if a friend was telling me this, what may I say?) and sympathic questions (e.g., how would I feel in this situation?) [22,42].

Narrative master plots that include acceptance as part of their plot structure are the most stable types [22]. The narrative master plots that illustrate an inability to accept what has happened or inability to see what is possible in the future may benefit most from storytelling interventions. Review evidence has identified that positive psycho-emotional changes are possible following storytelling interventions for people with H&NC [21]. This likely occurs by sharing illness narrative master plots that include acceptance or enable an individual to look differently at their situation [22]. Recent evidence supports the idea that people may be more likely to be persuaded to change attitudes and behaviours by those they consider ‘similar’ to themselves [43,44]. Further research is required to establish this.

### 4.1. Implications

This work provides several generic implications: People with H&NC do not have to be trained on how to tell narratives and it is a medium through which the perspective of a patient can be portrayed. The representations of illness narrative master plots are beyond simplistic labels, or the negative aspects of the experience. For instance, identifying a heroic/admirable /victim role and response during the treatment and recovery.The narrative master plots provide evidence of common narrative plots told by patients. They provide insight to the meaning behind the plot and give detail into how people with H&NC could use adaptation and coping, may interact with others and refer to information given to them.Listening for and noticing different narrative master plots may improve HCPs’ understanding of people in their care and could reduce some of the inherent stresses associated with working in a challenging environment of H&NC because the HCP recognises some of the presentations and can adjust their interaction to enhance empathy.Narrative master plots should not represent a static view of an individual with H&NC. Rather, people who listen to illness narratives of people with H&NC should recognise that the narrative master plots can change or be reworked.Narrative master plots can be documented using a brief five-question outcome measure [23]. The outcome measure results can be plotted to a model that documents the most important difficulty identified by the individual according to the hope that they have, as to whether it will change in the future (no hope/hope), their ability to accept what has happened, (from an inability to accept to an embracement of current circumstance) and what emotions are associated with the difficulty. HCPs can use this information to consider (a) which elements of psycho-emotional adaptation may be restricted and the need for psychological support and (b) the effectiveness of an intervention they may suggest or plan.

The current research has specific implications that relate to each narrative master plot identified;
People who tell the responsive and reflective narrative master plot often present it with a lack of emotional expression. HCPs may need to allow more time to hear and consider this narrative, identifying aspects of it that may allude to emotions through the use of metaphors and humour. Being able to clarify meaning to such expressions is important because the discussion will develop the description for the patient and support their reflection. For instance, an individual may say “*I feel like I’m a small boat on an stormy sea and my engine keeps stalling and I’m looking for a mechanic but everyone just hands me an instruction book which I won’t read*” as part of expressing an experience.People who tell a survive or die narrative often live in the moment and will want only the bare minimum of information. Whilst they will appreciate that they have to be informed of the current situation, they quickly defer to others and become overwhelmed, unable to absorb the information unless it is relevant to their current presentation. They would often place an onus on the HCP to recognise this and present information as it is needed rather than as a possibility.People who tell the frail narrative master plot require an opportunity to describe their current symptoms and the impact these have on their lives. There is a danger that HCPs close down such interactions too quickly. If the patients are not able to express the impact on them they can become irritated at being, in their terms, ignored, and they might repeatedly search for people who will listen to how the symptoms limit their lives, unable to consider adaptation until their current circumstances are acknowledged as real for them.People who tell the recovery narrative may not ask for help because they are stoical unless they perceive that the HCP team are not too busy or overburdened. They will be alert to HCPs’ non-verbal signals of being busy or tired themselves and may opt not to put them or the overstretched service under more pressure. The onus on the HCP is to attend fully to the patient and not be rushed by external factors. If the individual senses the HCP is rushed, the danger is that the patient will not discuss what matters to them.People who tell a personal project narrative master plot are at risk of being labelled as argumentative or challenging by the HCP. They are likely to be dismissive of members of the team who do not answer questions comprehensively. They might also use jargon that they have picked up from the literature without having a complete understanding of it. HCPs should take time to clarify information and summarise situations which this group see as a useful function. It does require time and a clinician who has detailed clinical knowledge of the situation to have a dialogue with them—if this cannot happen, they can become irritated that they are not being listened to or taken seriously. Simply clarifying whether an HCP has provided enough detail may help this.

### 4.2. Limitations

The number of participants in the current study is small. The interviews represent people who were willing to talk about their experiences and who engaged with the study. It could be that this is a group that has self-selected into the study, which means that other patients are represented poorly. We were not able to determine how confounding variables or key demographics may have impacted or interacted with the narrative master plot told by individuals. It is important to emphasise that there might be other plots that as yet are not well represented. This research was not able to illustrate how or why plots change and this should be a focus of future research. 

## 5. Conclusions

The current research provides a unique insight into common illness narrative master plots told by individuals with H&NC. These unique narrative plots provide an understanding of how a patient perceives their experiences. This has important implications for HCPs who may need to listen and respond to the narratives. There needs to be further research that understands other stakeholder’s responses to these narrative types and which acknowledges the role that carers might have on the experiences of patients.

## Figures and Tables

**Table 1 behavsci-09-00110-t001:** The demographic details for participants in the study-aligned to narrative master plots told.

Participant Number	Narrative Master Plot	Gender	Age at Time of Study (years)	Time since Diagnosis (years and months)	Site of Cancer	Treatment
**1**	1	M	55.1	1.1	Floor of mouth	Surgery + chemo-radiotherapy
**2**	1	M	46.0	1.1	Retro-molar + soft palate	Surgery + radiotherapy
**6**	1	M	61.1	1.3	Tonsil	chemo-radiotherapy
**12**	1	F	37.0	1.0	Buccal + mandible	Surgery + chemo-radiotherapy
**13**	1	F	51.1	1.0	Tongue base	chemo-radiotherapy + neck dissection
**14**	1	M	62.1	1.0	Tonsil	Surgery + radiotherapy
**17**	1	M	64.1	1.0	Laryngeal	Surgery + chemo-radiotherapy
**9**	2	F	65.0	1.0	Tongue base	Bilateral neck dissection + chemo-radiotherapy
**16**	2	M	55.1	1.0	Tonsil	Surgery + radiotherapy
**4**	3	F	65.1	1.3	Tonsil	chemo-radiotherapy
**5**	3	F	55.0	1.3	Tonsil	Surgery + chemo-radiotherapy
**18**	3	M	60.0	1.0	Oral pharyngeal	Surgery + radiotherapy
**3**	4	M	77.1	1.0	Oral tongue	Surgery
**7**	4	M	61.1	1.0	Laryngeal	Surgery + radiotherapy
**11**	4	F	58.1	1.1	Laryngeal	Surgery
**10**	5	M	40.0	1.0	Mandible	Surgery + chemo-radiotherapy
**15**	5	M	61.1	1.0	Floor of mouth	Surgery + radiotherapy
***8**	-	M	69.1	1.0	Maxilla	Surgery

Note: Master plot 1 = The response and reflective narrative; Master plot 2 = The frail narrative; Master plot 3 = The recovery narrative; Master plot 4 = The survive or die narrative; Factor 5 = The personal project narrative. * = No narrative type identified.

**Table 2 behavsci-09-00110-t002:** Characteristics of the teller and dangers and strengths of the master plots.

Narrative Master Plot	Noticeable Characteristics of the Teller	Dangers of the Master Plot	Strengths of the Master Plot
The responsive and reflective narrative	Identified as pragmatic about life. Ready to take action and embrace uncertainty with little evidence of fear or worry. The story did not appear frequently told because the teller has no apparent need to reveal their story to benefit themselves. The noticeable stability of the story appears to be developed from and through previous experiences they or their family have had. The tellers’ own mortality will have been considered prior to this diagnosis. No regrets appear around treatment. The teller appears to have insight that people respond differently, which means they will not offer advice readily. The teller has an innate confidence that they will cope.	The teller may be too ready to give up or accept an inevitable outcome. The teller does not express emotions easily. They may appear self-contained because they are skilled at telling a story that can rationalise out emotions. The teller appears to rely on finding the right people to relate to. Needs time to reflect and needs to be able to express views in ways that might seem or appear understated but are exposing for them. Decisions are made based on information that may be limited to what they know. They may not seek more information and will not have read information they are given or researched the web, which they believe will be too general.	The teller may appear to be able to articulate information objectively and adjust the amount to the role of the listener. The grief process may be limited for the teller and carried out internally with little need to refer to others. The teller will be responsive to the idea of treatment and key interactions. Often being able to recite information told to them by the HCP. The teller is pragmatic and not likely to be isolated because they are resourceful and able to create relationships if they want them.
The frail narrative	The teller can reflect on the toll the illness has had especially on their energy and reduced social roles. The teller identifies a lack of energy and their review of the losses experienced appears to limit engagement with society. If given the arena and opportunity, the teller will describe the symptoms in detail. For the teller, there is transiently a regret that they opted for treatment. They seek being at pre-morbid levels and their current position underlines the gap between now and then.	The teller is able to identify reasons why they cannot carry out more activities or engage in social occasions. There appears to have heavy reliance on HCPs for decisions and support. The teller might become isolated/lonely from their family who may ‘misunderstand them’ in their terms. The teller may respond with irritation to others’ reactions to their physical changes, which limits their social interactions. They can appear frustrated by systems and people that use up their emotions and energy which are scarce.	The teller can express a fixed position regarding their limitations and energy resources. The teller is often identified as having read information they are given by HCPs and understood it or will ask questions if they do not. They trust HCPs who they believe treat them as individuals who are valued
The recovery narrative	The teller represents a proactive group, and to do nothing was never an option. Belief that they can respond and understand the reality of the symptoms in an effective way. The teller often relies on others. Emotional language is often used to describe present circumstances. The tellers want people to listen and are sensitive to a perception of being overlooked. Whilst the tellers could recall other family members having a cancer diagnosis, they had not contemplated own death.	Stoic—will put up with symptoms. The teller may not be self-sufficient, needing relationships with HCPs. The teller could be sensitive to non-verbal cues that are not at the level they come to expect. The teller does not prepare for consultation or seek further clarification at an appointment. They can seek and obtain a paternalistic relationship with HCPs. Related to this, they may need several reviews and information repeated/broken down. The teller is likely not be opinionated in front of people they believe are in positions of power.	Integrates well into own social network, unlikely to be isolated and is proactive. May have no perception or sense of being stared at by others. The teller can remember key conversations and listen intently but is unlikely to read information. They can recognise times of vulnerability and can express them in emotional terms; it is easy for HCPs to know how they are because they will be honest.
The survive or die narrative	The teller is optimistic and lucky but precarious because if they learn of other people having a recurrence, they believe that this could be them. The teller lives in the moment and is reliant on others to enquire about future/treatment plans. They want to have minimal information. The teller typically would defer important decisions to HCPs with the view that ‘they know best’.	The teller places onus on HCPs to sum-up relevant info in order to make decisions. They are not active in their care but have others around them who often are. The teller appears not to process information received. Past experience may aid this.	Fatalistic and will not worry about what they cannot control. Optimistic. Expect others to help and acknowledge their need for others to help them. Not physically drained by the disease and treatment.
The personal project narrative	Independence is critical through the project that they are central to. The teller can identify and share that the recovery brings a sense of secondary gain and they are motivated by altruism. The teller has no regrets and life has more meaning.	The teller appears argumentative and could require a lot of time and require a high-level of clinical knowledge from HCPs. The teller may be dismissive of people who, for them, are ‘not up to the job’. The teller will test systems and people to determine whether they are. They discuss complex and abstract ideas using jargon; HCPs have to check that they have grasped this rather than having superficial knowledge. The teller may struggle with putting their lives in others’ hands, because of a lack of control. Technical jargon and behaving as professional can mask vulnerabilities that they might need to discuss.	Self-centred and self-motivated. Will learn through researching the literature and questioning HCPs. Will use data to demonstrate how they are. Will want to discuss detailed information during consultations. Will seek to be independent in care quickly.

Note: HCPs = Health Care Professionals.

**Table 3 behavsci-09-00110-t003:** Illustrating a detailed exploration of key features of the plot and tellers’ reaction of each narrative master plot.

Exploring the Plot (red) and the Teller (blue)	The Responsive and Reflective Narrative	The Frail Narrative	The Recovery Narrative	The Survive or Die Narrative	The Personal Project Narrative
**Choice in decision to be treated**	None	Had a choice and chose treatment	No doubts	No choice but to face it	No choice but an easy decision
**Regrets about decision**	None	On a bad day, yes	No regrets	No regrets	Cannot imagine thinking of regretting it. That thought is a ridiculous concept
**View of progress**	Recovery appears much slower than expected	Never appreciated the length of time needed. A time frame helps	End points encourage patient. They often take progress as a day at a time	Quick actually, but still a long time to experience	Amazing and very proud
**Interaction with HCPs and the information they are given.**	Stress the importance of being honest and do not give them false hope but keep them in their terms within a realistic framework	Reliant on the HCPs to have a good relationship with them and make the decision about how much information they can cope with. Read all the information and want ongoing discussion	No research or reading completed	Be honest, cannot absorb information until experience it	Need to read the information and discuss it, never trust the HCPs completely—their systems are suspect and could be better, and want to discuss this aspect of the care rather than their own needs
**Goals of the story**	Cure	Cope with the day to day symptoms	Cure important, make the goals achievable	Cure but do not face it alone	Understand every intricacy and how it impacts on them
**Perspective on mortality and/or recurrence**	Considered own death prior to diagnosis	Symptoms might be recurrence	Any symptom could be recurrence	Never happen to me to any symptom could be cancer	Never thought it could be them
**Expressions relating to physical, psychological and spiritual well-being**	Not vulnerable physically Do things in their own way and on their own terms. Do not face it alone	Exhausted physically and emotionally more irritated	Ill and now recovered—a long timescale	Could be impulsive, knew people avoids them but that is their prerogative	Do not need to conform to society in ways they used to. They often note an inner strength, never identified before diagnosis
**Adaptation, Recovery and hopes**	Isolation prevents ability to share narrative	Plod on and try and cope, personal isolation and others cannot understand what has happened to them	Life will never be the same again but deal with it	Life beyond the diagnosis, but fearful when others discuss possibility of cancer, that it could be them	Embody recovery and, against the odds, very hopeful that their experience is something others might benefit from
**Characteristics of story**	Pragmatic and reflective	Endure, but know the intricacies of treatment and recovery	Could not understand the treatment until the reality was being lived	Got away with it but could be next time	Keep control for sake of family and future patients

Note: HCPs = Health Care Professionals.

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
