# Peer review of "A Qualitative Study Examining the Illness Narrative Master Plots of People with Head and Neck Cancer"

_behavsci, 2019, doi:10.3390/bs9100110_

Round 1

Reviewer 1 Report

Thank you for the opportunity to review this well-written and innovative manuscript which describes common plots of illness narratives for people treated for head and neck cancers derived through qualitative methods. Overall, this manuscript describes an innovative approach to understanding how people with head and neck cancer may describe their illness and treatment experience and suggests that understanding these new narratives may enhance therapeutic relationships between patients and healthcare providers. I note three concerns:

Please provide further detail about the Q-sort, including a brief description of the methodology and summary of the findings of the Q-sort that was conducted prior to the individual interviews. The use of the Q-sort findings in informing the interviews is confusing and needs clarification and further description. Please provide a thorough discussion of how the author anticipate healthcare providers might use the presented findings, including the narratives, in clinical practice. Provide more specifics on how the provider might identify the narrative of a patient and how they might respond differently to a particular patient given a particular master-plot. The authors suggest that master-plots may be dynamic within a given patient. More discussion about this is needed.

Author Response

Reviewer One. 

R1

Thank you for the opportunity to review this well-written and innovative manuscript which describes common plots of illness narratives for people treated for head and neck cancers derived through qualitative methods. Overall, this manuscript describes an innovative approach to understanding how people with head and neck cancer may describe their illness and treatment experience and suggests that understanding these new narratives may enhance therapeutic relationships between patients and healthcare providers.

AS et al: Thank you for these comments and the comments below. We will send the editor a marked version of the manuscript so that you can see where changes have occurred.

R1

I note three concerns: Please provide further detail about the Q-sort, including a brief description of the methodology and summary of the findings of the Q-sort that was conducted prior to the individual interviews. The use of the Q-sort findings in informing the interviews is confusing and needs clarification and further description.

AS et al: Thank you for these comments. We have given further details of the Q-sort and included a description of the methodology and summary of the findings within section 2.4.

R1

Please provide a thorough discussion of how the author anticipate healthcare providers might use the presented findings, including the narratives, in clinical practice.

AS et al: We have provided this information now and rewritten the discussion to account for this comment.

R1

Provide more specifics on how the provider might identify the narrative of a patient and how they might respond differently to a particular patient given a particular master-plot.

AS et al: we have updated this information including identification and responses to particular plots.

R1

The authors suggest that master-plots may be dynamic within a given patient. More discussion about this is needed. 

AS et al: we have updated this considering the psychological aspects which make them dynamic.

Reviewer 2 Report

The article entitled “A qualitative study examining the illness narrative master-plots of people with Head and Neck Cancer” responds to the need to understand the so called “master plot” (a commonly recognisable story plot relating to the experience of illness), in 18 patients treated for Head and Neck Cancer. 5 master-plots were identified: (1) The responsive and reflective narrative, (2) The frail narrative, (3) The recovery narrative, (4) The survive or die narrative and (5) The personal project narrative.

The identification of common narrative master plots of patients with Head and Neck Cancer give some insight into how these patients could use the mechanisms of adaptation and coping. Furthermore, the narrative master plots of these patients can change or be reworked over time.

Although the number of patients is small, this is an interesting study, conducted with a rigorous method. The text clear and easy to read and the Discussion is quite balanced. Overall, this study represents an excellent starting point for further studies.

I suggest adding some considerations in the Introduction section regarding the unsatisfactory 5-year survival rate of Head and Neck Cancer patients, such as Oral Squamous Cell Carcinoma, despite the continuous search for new prognostic and predictive factors [1].

[1] Mascitti M, et al. American Joint Committee on Cancer staging system 7th edition versus 8th edition: any improvement for patients with squamous cell carcinoma of the tongue? Oral Surg Oral Med Oral Pathol Oral Radiol. 2018 Nov;126(5):415-23.

Author Response

R2

The article entitled “A qualitative study examining the illness narrative master-plots of people with Head and Neck Cancer” responds to the need to understand the so called “master plot” (a commonly recognisable story plot relating to the experience of illness), in 18 patients treated for Head and Neck Cancer. 5 master-plots were identified: (1) The responsive and reflective narrative, (2) The frail narrative, (3) The recovery narrative, (4) The survive or die narrative and (5) The personal project narrative.

The identification of common narrative master plots of patients with Head and Neck Cancer give some insight into how these patients could use the mechanisms of adaptation and coping. Furthermore, the narrative master plots of these patients can change or be reworked over time.

Although the number of patients is small, this is an interesting study, conducted with a rigorous method. The text clear and easy to read and the Discussion is quite balanced. Overall, this study represents an excellent starting point for further studies.

AS et al: thank you for this.

I suggest adding some considerations in the Introduction section regarding the unsatisfactory 5-year survival rate of Head and Neck Cancer patients, such as Oral Squamous Cell Carcinoma, despite the continuous search for new prognostic and predictive factors [1].

 [1] Mascitti M, et al. American Joint Committee on Cancer staging system 7th edition versus 8th edition: any improvement for patients with squamous cell carcinoma of the tongue? Oral Surg Oral Med Oral Pathol Oral Radiol. 2018 Nov;126(5):415-23.

 AS et al: this has been added.